# Inhibition of eIF6 Activity Reduces Hepatocellular Carcinoma Growth: An In Vivo and In Vitro Study

**DOI:** 10.3390/ijms23147720

**Published:** 2022-07-13

**Authors:** Alessandra Scagliola, Annarita Miluzio, Giada Mori, Sara Ricciardi, Stefania Oliveto, Nicola Manfrini, Stefano Biffo

**Affiliations:** 1National Institute of Molecular Genetics, Fondazione Romeo ed Enrica Invernizzi, Via Sforza 35, 20122 Milan, Italy; scagliola@ingm.org (A.S.); miluzio@ingm.org (A.M.); mori@ingm.org (G.M.); ricciardi@ingm.org (S.R.); oliveto@ingm.org (S.O.); manfrini@ingm.org (N.M.); 2Department of Biosciences, University of Milan, Via Celoria 26, 20133 Milan, Italy

**Keywords:** fatty acid synthesis, eIF4E, initiation, uORFs, HCC, eIF6-60S binding inhibitors

## Abstract

Nonalcoholic fatty liver disease (NAFLD) is characterized by the accumulation of lipids in the liver. Given the high prevalence of NAFLD, its evolution to nonalcoholic steatohepatitis (NASH) and hepatocellular carcinoma (HCC) is of global concern. Therapies for managing NASH-driven HCC can benefit from targeting factors that play a continuous role in NAFLD evolution to HCC. Recent work has shown that postprandial liver translation exacerbates lipid accumulation through the activity of a translation factor, eukaryotic initiation factor 6 (eIF6). Here, we test the effect of eIF6 inhibition on the progression of HCC. Mice heterozygous for eIF6 express half the level of eIF6 compared to wt mice and are resistant to the formation of HCC nodules upon exposure to a high fat/high sugar diet combined with liver damage. Histology showed that nodules in eIF6 het mice were smaller with reduced proliferation compared to wt nodules. By using an in vitro model of human HCC, we confirm that eIF6 depletion reduces the growth of HCC spheroids. We also tested three pharmacological inhibitors of eIF6 activity—eIFsixty-1, eIFsixty-4, and eIFsixty-6—and all three reduced eIF6 binding to 60S ribosomes and limited the growth of HCC spheroids. Thus, inhibition of eIF6 activity is feasible and limits HCC formation.

## 1. Introduction

Hepatocellular carcinoma (HCC) is the most common type of primary liver cancer. HCC occurs most often in people with chronic liver diseases. Cirrhosis caused by hepatitis B and hepatitis C infection, and alcohol abuse are well-established predisposing conditions for HCC [1]. More recently, NAFLD (nonalcoholic fatty liver disease) has emerged as a risk condition for HCC development. NAFLD is defined as the accumulation of lipids in the liver, in general associated with obesity and metabolic syndromes. Nonalcoholic steatohepatitis (NASH), the progressive form of NAFLD, is another risk factor for the onset of HCC. The annual progression rate of cirrhotic NASH to HCC ranges from 2.4% to 12.8% [2]. Interestingly, some causative factors and clinical or molecular features may differentiate NASH-related HCC from other forms of HCC. Compared to viral- or alcohol- associated HCC, NASH-HCC is more frequent in non-cirrhotic livers and in older patients with metabolic diseases. A molecular analysis of a large cohort of NASH-HCC patients has shown that it displays unique molecular features, including higher and specific rates of somatic mutations when compared to other etiologies [3]. These data suggest that NASH-driven HCC has unique features and may be sensitive to the manipulation of specific metabolic signaling pathways, such as lipogenesis.

Translational control can be defined as the series of events that lead to the selection of specific mRNAs to be translated by the ribosomal machinery [4]. Notably, several studies have shown a poor correlation between mRNA and protein levels, indicating that translation is the main mechanism that regulates the amount of synthesized protein [5]. Translation is under the control of eukaryotic initiation factors (eIFs). These eIFs are regulated by signaling pathways and stimulate the translation of specific mRNAs by interacting with 5′UTR sequences of mRNAs [6,7]. In cancer, nutrient signaling pathways strongly converge on two initiation factors, eIF4E and eIF6. eIF4E positively regulates the translation of oncogenic mRNAs downstream of mTORC1 and ERK [8]. eIF6 is rate limiting for the growth of several cancer cells with active RAS or Myc amplification [9]. Notably, inhibition of either eIF4E [10] or eIF6 [11] results in the reduction of oncogenesis and tumor growth. These findings have been extended to multiple cellular and animal models [12]. In agreement with the growing relevance of translational machinery in cancer, pharmacological strategies targeting translation factors are being developed [13].

In the specific context of liver biology and HCC, the role of translation may be of particular importance. First, the liver is the organ with the highest levels of protein synthesis [14]. Second, translation factors like eIF4E are essential for HCC development [15,16]. Third, ribosome-associated factors have prognostic significance in HCC; for instance, RACK1 ribosomal protein, an intriguing receptor for activated kinase C that binds 40S subunits close to the mRNA exit channel [17,18,19,20,21]. RACK1 may additionally regulate eIF6 activity [22]. Notably, eIF6 itself is also associated with HCC progression [23,24]. Fourth, increased lipogenesis promotes the development of HCC [25] and translation factor activity robustly intersects with lipid metabolism [26]. Taken together, current evidence suggests that translation factors have a continuous role during the evolution from NAFLD to HCC.

eIF6 activity has specific features that may impact on the evolution of HCC. eIF6 increases the translation of lipogenic and oncogenic transcription factors downstream of several signaling cascades [27]. Consistent with this finding, we have recently reported that eIF6 heterozygous mice are protected from the evolution of NAFLD [28]. Some short upstream open reading frames known as 5′ uORFs inhibit the translation of the downstream main open reading frame (ORF). Mechanistically, eIF6 increases the synthesis of lipogenic transcription factors through the stimulation of reinitiation of translation of ORFs that reside downstream of an uORF [29]. Therapeutic targeting of eIF6 is facilitated by the observation that eIF6 inhibition of Myc-induced lymphomagenesis more than doubles survival without overt negative effects [11], suggesting the possibility of a therapeutic window for eIF6 inhibition. The molecular action of eIF6 on reinitiation is exerted through its regulated binding to 60S ribosomal subunits [22]. In this context, we have identified three compounds, eIFsixty-1, eIFsixty-4, and eIFsixty-6 that inhibit the binding of eIF6 to 60S and the translation of lipogenic enzymes [30]. However, the extent by which eIF6 genetic or pharmacological inhibition impairs the formation and growth of HCC is still not fully addressed. Here, we first evaluated the extent of protection eIF6 heterozygosity provided in a mouse model of NAFLD-HCC evolution. Next, we employed an in vitro model of HCC spheroids [31] to evaluate the effect of both genetic and pharmacological inhibition of eIF6. Our data show that both genetic and pharmacological inhibition of eIF6 limits HCC tumorigenesis and growth. Thus, we provide proof-of-principle that the inhibition of the translational machinery driven by eIF6 is a useful approach for liver cancer treatment without any overt negative effects.

## 2. Results

### 2.1. eIF6 Depletion In Vivo Delays HCC Nodules Formation and Growth without Overt Negative Side Effects

eIF6 heterozygous (het) mice have 50% eIF6 protein levels compared to wt mice [9]. We have previously found that eIF6 het mice are protected from a high fat diet (HFD) [28]. Here, we administered mice a pro-carcinogenic reagent (DEN) at 14 days of age, and after weaning, they were subjected to a high fat diet combined with high sugar water and weekly injections of CCl_4_ (Figure 1A). We then analyzed at early and late time points (six weeks and eighteen weeks, respectively) the onset of HCC nodules in both wt and het eIF6 mice (Figure 1B,C). Six weeks after the diet, wt mice had almost 100% nodules of 2–6 mm size. In contrast, het mice still had a large majority of very small nodules, less than 2 mm. At eighteen weeks after the diet, eIF6 het mice still had a large majority of nodules of less than 2 mm, whereas wt mice had developed larger nodules, up to 1 cm in size (Figure 1C). These data suggest that a 50% reduction in eIF6 levels strongly inhibits nodule growth. We assessed the functionality of the liver by measuring aspartate aminotransferase (AST) and alanine aminotransferase (ALT), since ALT and AST increase upon liver damage [32]. Het eIF6 mice had lower levels of ALT and AST at all time-points (Figure 1D) compared to wt mice, indicating that eIF6 heterozygosity reduces liver damage.

We examined the morphological aspect of liver sections. At the early time point, eIF6 het mice had less fibrosis, as shown by Sirius Red staining, as well as less steatosis and reduced eIF6 levels (Figure 2A). Interestingly, the livers of wt mice had higher levels of Ki-67 staining, a marker of proliferating cells (Figure 2A). Quantitation of fibrotic areas and proliferating cells confirmed that eIF6 het mice were less fibrotic (Figure 2B) with less proliferative areas than wt mice (Figure 2C). eIF6 levels in het mice were reduced, as expected (Figure 2D).

At later time points, eIF6 het mice showed a clear delay in the formation and growth of tumors. First, eIF6 het now had more fibrosis than wt mice, as shown by Sirius Red staining and quantitation (Figure 3A,B). Second, eIF6 het mice presented less Ki-67 rich areas (Figure 3A–C). These data indicate that the progression from fibrosis to cancerous lesion occurs earlier in wt mice. eIF6 levels are reduced in het mice, as expected (Figure 3D). Alpha fetoprotein (AFP) is a liver tumor marker [33] and wt mice had high levels of AFP in comparison to eIF6 het (Figure 3D). In conclusion, wt mice develop HCC before het mice and with higher proliferation rates.

### 2.2. eIF6 Depletion In Vitro Consistently Reduces HCC Growth in an Established Human 3D Tumor Model

The results obtained in vivo suggest that a modest inhibition of eIF6 activity is sufficient to inhibit tumor growth. To further address this issue, we used Huh7.5 tumor spheroids (Figure 4A). In this model, cells are grown in a hanging drop adherent on plastic. For eIF6 inhibition, we used two separate shRNA for eIF6. Both were effective in the downregulation of eIF6 protein, as tested in 2D-cultures (Figure 4B). Next, 3D-cultures were grown up to 7 days. Under these conditions, we found that each eIF6 shRNA inhibited the growth of the tumor spheroid (Figure 4C). We checked the eIF6 protein levels in the spheroids and found it was detectable, suggesting there was an evolutionary pressure for re-expression of eIF6 during the growth of the HCC spheroid (Figure 4D and Appendix A). This said, the reduction in the expression of eIF6 and spheroid growth was matched by reduced AFP levels. Thus, the in vitro data for Huh7.5 cells is in line with our in vivo results and suggest that inhibition of eIF6 expression reduces HCC growth. When we quantitated HCC growth, silencing eIF6 resulted in 30–50% inhibition with total penetrance (Figure 4E).

### 2.3. eIF6 Pharmacological Inhibition Reduces the Growth of HCC Spheroids

Recently, we identified three inhibitors of eIF6 binding to the 60S subunit. All three reduce the translation of lipogenic enzymes [28,30,34]. The inhibitors will be referred to as eIFsixty-1, eIFsixty-4 and eIFsixty-6. We first tested their effect on eIF6 binding to 60S in Huh 7.5 cells, and then, whether they inhibited HCC growth.

For this experiment, we used a variation of the iRIA protocol [34]. We purified 60S Arthemia salina ribosomes and eIF6 and validated their functionality (Appendix A). Next, we immobilized 50 micrograms of Huh 7.5 protein extract in a ELISA microwell and incubated it with recombinant eIF6, either in the presence or absence of each eIF6 inhibitor. The results show that each inhibitor, when used at its IC_50_ concentration (1.4 µM for eIFsixty-1; 5 μM for eIFsixty-4; 1 µM for eIFsixty-6) [30] reduced the binding of eIF6 to 60S (Figure 5A). Positive controls (pure ribosomes in the absence of inhibitors) and negative controls (absence of eIF6 in the reaction) are shown for comparison (Figure 5A). We then tested the effect of each inhibitor on the growth of Huh7.5 cells in 2-D culture at 24 h and 48 h. The data show that eIFsixty-4 was the only compound that was rapidly cytostatic at 24 h (Figure 5B), but there was some recovery at 48 h due to reduced stability (Figure 5B, right). A minor effect of eIFsixty-1 and eIFsixty-6 on proliferation was evident at 48 h (Figure 5B).

We then checked the effect of each inhibitor on the initiation of translation in vitro. Polysomal profiles allow an estimation of translational initiation by determining the ratio of polysomes to 80S subunits [19]. An increase in the 80S peak relative to the polysome peaks is known to be due to a decrease in the initiation of translation. In this experiment, eIFsixty-1, eIFsixty-4 and eIFsixty-6 all induced an increase in the 80S peak, thus confirming that they can block the initiation of translation. eIFsixty-4 also induced a reduction in the polysome peaks, indicating a stronger inhibitory effect on translational initiation (Figure 5C). None of the inhibitors had an effect on absolute eIF6 levels (Figure 5D). Taken together, the data demonstrate that eIFsixty-i (eIFsixty-1, eIFsixty-4, eIFsixty-6) are able to reduce the binding of eIF6 to 60S and the initiation of translation. However, in 2D-cultures, only eIFsixty-4 seems to reduce the growth of Huh 7.5 cells.

Last, we analyzed the effect of eIFsixty-1, eIFsixty-4 and eIFsixty-6 on the growth of Huh7.5 3D spheroids. The three compounds were administered to cells at their IC_50_ concentration. eIFsixty-4 was highly toxic to Huh 7.5 cells during 3D spheroid generation and totally abrogated its formation (Figure 6A), whereas eIFsixty-1 and eIFsixty-6 reduced spheroid growth (Figure 6B,C) by approximately 50%. None of the three compounds had an effect on eIF6 levels (Figure 6C), but each reduced the levels of AFP (Figure 6C). In conclusion, administration of eIFsixty-i inhibitors mimicked the effects of eIF6 genetic inhibition in the 3D spheroid assay by inhibiting the binding of eIF6 to 60S ribosomal subunits. The observed differences in their relative activities are consistent with differential stability and different sites of binding to the eIF6-60S surface.

## 3. Discussion

In this work, we asked whether three recently identified antagonists of the binding of eIF6 to the 60S ribosome, eIFsixty-1, eIFsixty-4 and eIFsixty-6, have an inhibitory activity on HCC spheroids that is comparable to genetic inhibition of eIF6. We demonstrate that all three eIF6 inhibitors delay to some extent HCC growth. We will discuss several issues.

The primary limitation of our study, for all inhibitors, is the absence of in vivo testing, which is greatly needed to define both their antitumorigenic potential as well as the extent of their potential side effects. At present, only one of these compounds can be easily tested, eIFsixty-1, which is a known antibiotic, clofazimine. It has been previously used in patients, but a number of adverse and toxic effects have been reported, and the drug is practically not employed anymore. Its mechanism of action remains largely debated. It is curious that clofazimine has been proposed as an anti-cancer agent acting that interferes with the Wnt pathway [35] given the well-known capability of eIF6 to enforce Wnt signaling [36]. In this context, studies that analyze in detail the effects of clofazimine on eIF6 are under way. More interestingly, eIFsixty-4 and eIFsixty-6 are novel hits. eIFsixty-4 was detected once in a virtual screening for arginine N-methyltransferase inhibitors and found to have an IC50 activity of ≅30 μM in biochemical assays [37], which is higher than the IC50 for the binding of eIF6 to 60S (≅5 μM). The most promising eIF6 inhibitor is eIFsixty-6 because it is not toxic in vitro and has a peculiar biochemical activity with features consistent with RNA binding. Currently, some limitations for progressing to in vivo testing have to be considered. eIFsixty-4 and eIFsixty-6 have large structures that are not amenable to extensive lead optimization, and they do not have solubility properties that will allow sufficient concentrations to be achieved in vivo. Furthermore, the synthesis of a large amount of newly-synthetized compounds requires extensive funding. Within these limitations, however, in vivo studies with eIFsixty-6 will be prioritized.

On the mechanistic side, we currently do not know whether eIF6 inhibitors bind 60S ribosomal subunits or eIF6 itself. It is somewhat expected that binding 60S subunits may have a more dramatic effect than binding eIF6, because the inter-subunit surface of 60S where eIF6 binds [38] is necessary for its association with the 40S subunit and subsequent translation. It is likely that the dramatic effect of eIFsixty-4 on HCC growth is linked to its capability to lock 60S subunits. This inhibitor totally blocks the binding of eIF6 to Arthemia salina ribosomes (Appendix A), suggesting that it may bind different 60S conformations. Finally, its effect is rapid, but less prolonged, thus suggesting instability. On the contrary, eIFsixty-1 and eIFsixty-6 have a milder effect on translation, which closely resembles eIF6 downregulation, suggesting that they might bind eIF6 itself. A total block of eIF6 activity is expected to be lethal [9]. In theory, long-term treatment with eIFsixty-1 and eIFsixty-6 may completely block eIF6-60S binding, with dramatic effects on 60S maturation, its cytoplasmic availability and consequently, mRNA translation. On the other hand, this is an unlikely scenario, given that: (i) these compounds could act on specific conformations of the free eIF6 protein and/or the eIF6-60S complex; (ii) a small amount of eIF6 protein (20%) is sufficient for ribosome biogenesis and cell survival [9]. We therefore think that complete pharmacological targeting of eIF6 activity will be difficult to achieve. These issues deserve, however, further analysis.

De novo lipid synthesis plays a dual role in NAFLD-related HCC, namely, both in the onset of HCC and in its progression. In cancer progression, it is known that FASN is a central enzyme that catalyzes a committed step in fatty acid synthesis [39]. Stabilization of FASN is central in the process of HCC evolution and its progression [40]. FASN silencing impairs HCC carcinogenesis in Akt-overexpressing mice [41]. Several studies have shown that up to 50% of patients with NAFLD-related HCC had no clinical or histological evidence of cirrhosis [42]. These observations imply that when the onset of HCC is driven by NAFLD, the obvious cellular insult is the accumulation of lipids due to the combination of reduced fatty acid oxidation and increased fatty acid synthesis. For this reason, the clear and coordinated effects of the inhibition of eIF6 activity are not surprising, but rather, expected. eIF6 translational activity is a central regulator of tumorigenesis and tumor growth because of its capability to upregulate fatty acid accumulation at the translational level. In the liver, the translational activity of eIF6 leads to a strong increase in FASN levels, whereas its inhibition rapidly reduces FASN levels [29]. We have proposed in the past that eIF6 downstream of insulin receptor activation increases both global translation and the specific translation of lipogenic enzymes [26,29]. The net result is the amplification of a lipidogenic circuit of energy storage during high nutrient levels, an essential evolutionary adaptation to fluctuating levels of nutrients, that we named metabolic learning. It is thus expected that a modest inhibition of eIF6 activity will have an impact on NAFLD evolution and HCC. Strikingly, we have recently observed that eIF6 inhibition also causes an indirect increase of fatty acid oxidation, since it maintains translational control that operates through different signaling pathways [28]. Here, we have extended these observations by showing that the growth of HCC is impaired by eIF6 inhibition and that eIF6 inhibitors behave as predicted, delaying the growth of HCC spheroids.

The future of pharmacological targeting of eIF6 has so far been hampered by a misconception of its lack of specificity and feasibility. These cautions are not justified since the biochemical activity of eIF6 on the 60S ribosomes is highly specific [43] and only a minimal level of eIF6 is necessary for survival and viability of adult cells [9]. As a master regulator of lipid metabolism at the translational level, the activity of eIF6 is meant to adapt metabolism to nutrient availability, so we expect that its modulation will lead to clear physiological effects.

## 4. Materials and Methods

### 4.1. Mice

eIF6^+/+^ and eIF6^+/−^ transgenic mice were generated as previously described [9]. For this study, a cohort of 15-day-old mice (*n* = 9 for eIF6^+/+^ and *n* = 7 for eIF6^+/−^) were intraperitoneally (i.p.) injected with diethylnitrosamine (DEN, dissolved in PBS, 100 mg/kg body weight) as a carcinogenic reagent [32]. Two weeks later, the mice were fed a HF diet (Research Diet D12451, containing 45 kcal% fat) and a high sugar water solution (23.1 g/L D-fructose, Sigma-Aldrich cat. No. F0127; 18.9 g/L D-glucose, Sigma-Aldrich cat. No. G8270). This diet regimen was started simultaneously with weekly i.p. injections of CCl_4_ (0.2 μL/g of BW; Sigma-Aldrich cat. No. 289116, St. Louis, MO, USA). To study the progression from NAFLD to HCC in this model, the mice were sacrificed at two different time points. Precisely, three eIF6^+/+^ and eIF6^+/−^ mice were sacrificed after 6 weeks of HFD and CCl_4_-treatment for the early time point, and six eIF6^+/+^ mice and four eIF6^+/−^ mice were sacrificed after 18 weeks for the late time point. At sacrifice, blood and liver tissues were collected for biochemical assays and the evaluation of nodules, respectively. Liver tissues were then processed for histological and further biochemical analysis.

All mice were maintained under specific and opportunistic pathogen-free conditions and all experiments involving animals were performed in accordance with the Ethical Committee of San Raffaele and experimental protocols approved by national regulators (IACUC n.688).

### 4.2. Biochemical Analysis

To address hepatic functionality, sera samples collected from all considered mice were used. Alanine aminotransferase (ALT) and aspartate aminotransferase (AST) activities were measured using two different commercial kits (cat. No. MAK052 and MAK055 Sigma-Aldrich, respectively) according to manufacturer’s instructions. Western Blotting analysis were performed on protein extracts obtained from both livers and cells using RIPA buffer (10 mM Tris-HCl, ph 7.4, 1% sodium deoxycholate, 1% Triton X-100. 0,1% SDS, 150 mM NaCl and 1 mM EDTA, pH 8.0). The following antibodies were used: rabbit polyclonal antibody against eIF6 [44] and mouse monoclonal antibodies against AFP (Santa Cruz) and β-Actin (Sigma-Aldrich).

### 4.3. Histological Staining and Immunohistochemistry

Liver samples were recovered at early and late time points and processed for histological analysis. Briefly, 5 µm paraffin-embedded sections of livers were stained with hematoxylin and eosin (Sigma-Aldrich) for morphological analysis and Sirius Red for collagen fiber detection (Sigma-Aldrich) according to the manufacturer’s instructions. All samples were later treated with tris-EDTA buffer pH = 9 for 30′ at 96° for antigen retrieval and processed for immunohistochemical staining for eIF6 (rabbit polyclonal anti-eIF6, 1:200, o/n) and Ki-67 (rabbit polyclonal anti-Ki-67, 1:200, o/n) using Vectastain Elite ABC kits (DBA) according to the manufacturer’s instructions. For the quantification of Sirius Red area, red-stained collagen was selected and measured by ImageJ. Ki-67 positive cells were quantified using Nis-Elements V5.30 software.

### 4.4. iRIA

eIF6 recombinant protein was isolated from the BL21(DE3) E. Coli strain transfected with the pET23 vector and purified as described in [22].

The iRIA assay is based on spotting 60S ribosomes on a microwell plate and incubating them with labelled eIF6. The purification of the 60S subunit ribosomes was performed as previously described [34]. The rationale and the controls for the procedure are described in [30,34]. In this specific work, wells were coated with 3 pM of 60S Arthemia salina ribosomes overnight at 4 °C. The next day, the ribosome mixture was removed and the wells saturated with 5% BSA in 0.5% Tween-20-PBS for 30 min at 37 °C. Recombinant eIF6 was then added to a final concentration of 5 pM in a volume of 100 μL in 5% BSA, 0.5% Tween 20 in PBS and incubated for 1 h at RT. Following three washes with 0.5% Tween 20 in PBS, a monoclonal antibody against eIF6 [11] (clone 8D10, 0.2 μg/mL) was added and incubation at RT for 60 min. After washing as above, the reaction was developed in tetramethilbenzidine (TMB, cat. No. T0440 Sigma-Aldrich) for 3–5 min. Absorbance was read at 450 nm after inactivation with H_2_SO_4_ 2 M.

### 4.5. eIFsixty-i Compounds

The eIFsixty-i compounds were previously identified from a subset of the CNCCS collection library (c. 150,000 compounds; www.cnccs.it, accessed on 27 February 2017) and FDA/EMA approved drugs, as described in detail in [30]. In particular, eIFsixty-1 is a compound previously approved for treatment of leprosy (clofazimine, (E)-N,5-bis(4-chlorophenyl)-3-(isopropylimino)-3,5-dihydrophenazin-2-amine); eIFsixty-4 is 14-benzoylacenaphtho [1,2-d]benzo[4,5]thiazolo[3,2-a]pyridin-13-ium,); and eIFsixty-6 is a compound pertaining to a class of predicted RNA binders (5-(4-benzylpiperazin-1-yl)-2-(2-phenylcyclopropyl)oxazole-4-carbonitrile). For the present work, we used the same eIFsixty-i compounds newly synthetized on demand by Ambinter SARL, 45100 Orléans, France. Their activity was tested in biochemical assays before use.

### 4.6. Cell Cultures

HEK293T and Huh7.5 cells were cultured in DMEM medium supplemented with 10% FBS, 100 U/mL penicillin, 100 µg/mL streptomycin and 1% glutamine. For shRNA experiments, Huh7.5 cells were transduced with lentiviral vectors carrying a scramble shRNA or two eIF6-specific shRNAs [29]. Huh7.5 cells were treated for 72 h with eIF6ixty-i compounds at their IC50 concentration (eIFsixty-1, 1.4 µM; eIFsixty-4, 5 μM; eIFsixty-6, 1 µM). Cell spheroids were generated from Huh7.5 cell culture using a hanging drop method [45]. After seven days, spheroids were analyzed with a microscope (Widefield, Leica DMI6000, Mannheim, Germany). The viability of Huh7.5 cells upon eIFsixty-i compound treatment was measured with Cell Titer blue (Cell Titer-Blue^®^ Cell Viability Assay, Promega, Madison, WI, USA cat. No. G8080) using an Infinite F200 plate reader (Tecan, Tecan Group Ltd., Männedorf, Switzerland).

### 4.7. Polysomal Profiles

Polysomal profiles were run as previously described [46] with the following specific variations. Cellular extracts were lysed in 50 mM Tris-HCl pH = 7.8, 240 mM KCl, 10 mM MgSO_4_, 5 mM DTT, 250 mM sucrose, 2% Triton X-100, 90 µg/mL cycloheximide and 30 U/mL RNasin for 30 min at 4 °C. After clearing at 14,000 g for 30 min, nucleic acids were quantitated at OD^260^. An equivalent of 10 units of OD was then loaded on a 15–50% sucrose gradient and centrifuged at 4 °C in a SW41Ti Beckman rotor for 3 h 30 min at 39,000 rpm. Absorbance at 254 nm was recorded by BioLogic LP software (BioRad, Hercules, CA, USA).

## 5. Conclusions

eIF6 levels are rate limiting for the growth of HCC. Genetic inhibition of eIF6 or its inhibition through novel drugs that block eIF6 binding to 60S ribosomal subunits reduce HCC growth.

## Figures and Tables

**Figure 1 ijms-23-07720-f001:**
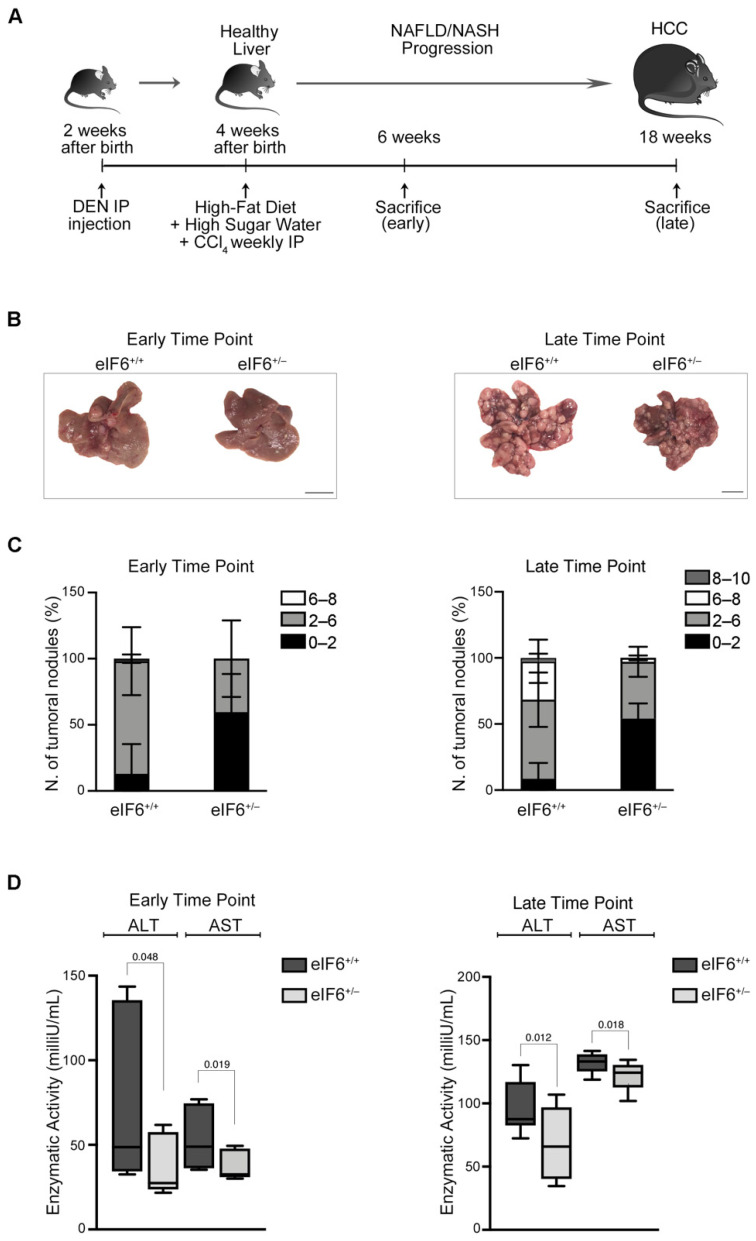
eIF6 haploinsufficiency reduces HCC nodule development and liver injury. (**A**) Schematic overview of NAFLD to HCC mouse model generation. (**B**) Representative images of eIF6^+/+^ and eIF6^+/−^ livers at two different time points. Scale bar = 1 cm. (**C**) Stacked bar charts representing the quantification of total surface tumors classified into four size ranges. Most nodules in eIF6^+/−^ livers were in the lower range. Early time point, *n* = 3 for each genotype; late time point, *n* = 6 eIF6^+/+^ mice and *n* = 4 eIF6^+/−^ mice. (**D**) Liver functionality evaluation: AST and ALT enzymatic activities were measured at the early (left) and late (right) time points. Data are represented as box-plots for *n* = 3 (early time point) and *n* = 4 (late time point) mice for each genotype. The bold black line indicates the median and the whiskers show the range from the minimum to maximum value. The results of two-tailed *t* tests are shown on each plot.

**Figure 2 ijms-23-07720-f002:**
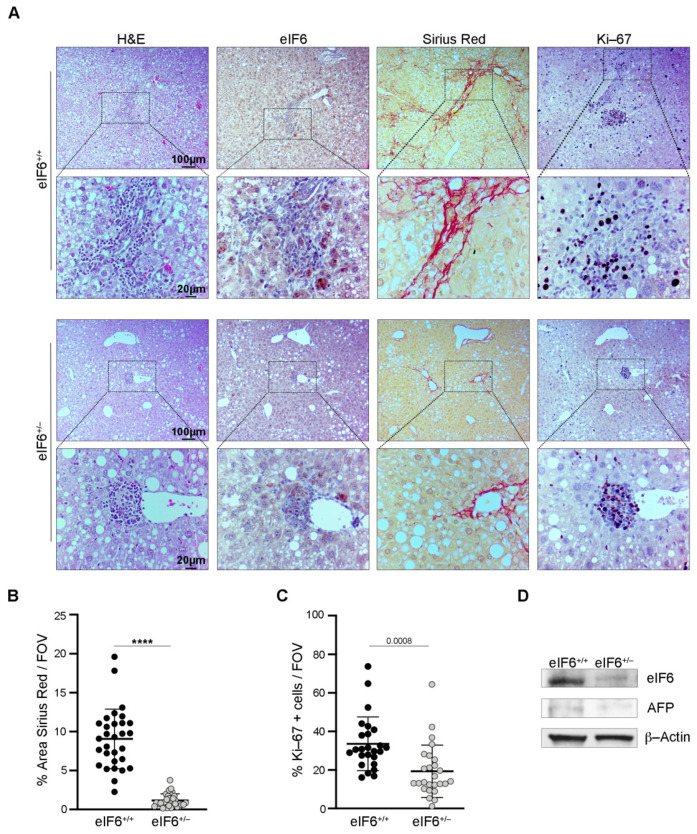
eIF6^+/−^ livers show less fibrotic areas and proliferating cells compared to wt livers. (**A**) Representative images of liver sections for the indicated stains and immunohistochemistry. Mice were sacrificed after 6 weeks of pre-carcinogenic treatment. Scale bars are indicated. (**B**) Quantification of the Sirius Red positive areas. Hepatic fibrosis is decreased in eIF6^+/−^ mice. *n* = 10 fields of view (FVO)/mouse. Data are represented as percentages and the bold black lines indicate the mean ± SD. Results of a two-tailed *t* test are shown. **** *p* value ≤ 0.0001. (**C**) Quantification of Ki-67 positive cells. The proliferation rate is decreased in eIF6^+/−^ mice. *n* = 10 fields of view (FVO)/mouse. Data are represented as percentages and the bold black lines indicate the mean ± SD. Results of a two-tailed *t* test are shown. (**D**) Representative Western blotting of eIF6^+/+^ and eIF6^+/−^ liver protein extracts. β-Actin was used as a loading control.

**Figure 3 ijms-23-07720-f003:**
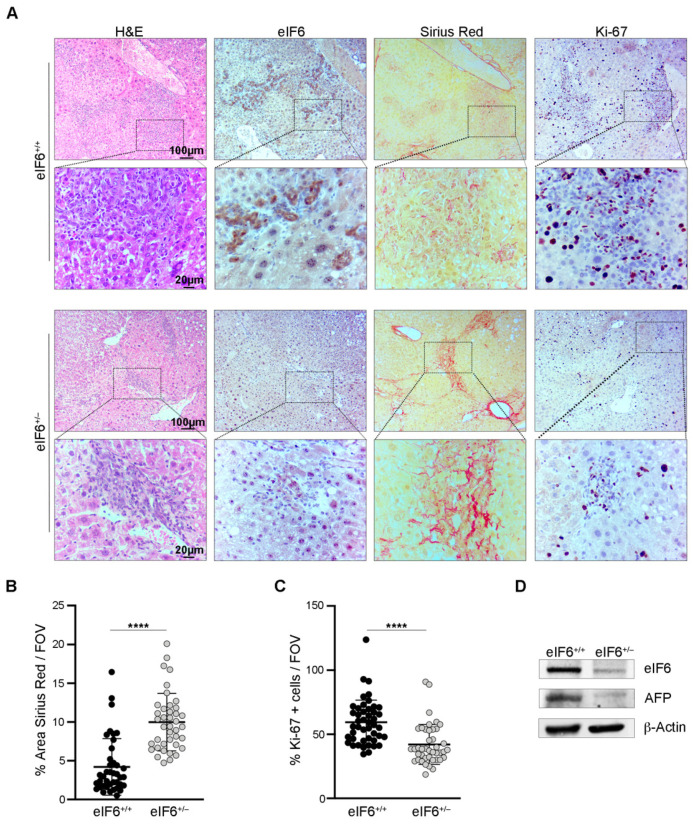
HCC tumor growth is reduced in eIF6^+/−^ livers. (**A**) Representative images of liver sections for the indicated stains and immunohistochemistry derived from mice sacrificed at the late time point (18 weeks). Scale bars are indicated. (**B**) Quantification of the Sirius Red positive areas. Hepatic fibrosis is increased in eIF6^+/−^ livers. *n* = 10 fields of view (FVO)/mouse. Data are represented as percentages and the bold black lines indicate the mean ± SD. Results of a two-tailed *t* test are shown. **** *p* value ≤ 0.0001. (**C**) Quantification of Ki-67 positive cells. Proliferation rate is decreased in eIF6^+/−^ livers. *n* = 10 fields of view (FVO)/mouse. Data are represented as percentages and the bold black lines indicate the mean ± SD. Results of a two-tailed *t* test are shown. **** *p* value ≤ 0.0001. (**D**) Representative Western blotting of eIF6^+/+^ and eIF6^+/−^ liver protein extracts. AFP protein levels are higher in eIF6^+/+^ livers. β-Actin was used as a loading control.

**Figure 4 ijms-23-07720-f004:**
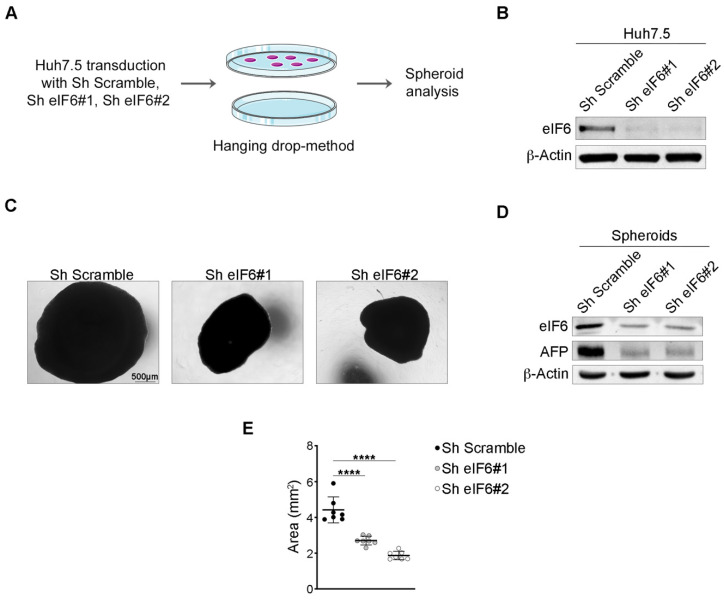
eIF6 depletion impairs HCC spheroid formation. (**A**) Outline of in vitro generation of a 3D-tumor model. (**B**) Representative Western blotting of eIF6 protein levels in Huh7.5 cells transduced with two ShRNA for eIF6 and one control (Sh Scramble). (**C**) Brightfield images of spheroids derived from Huh7.5 cells silenced for eIF6. Sh Scramble-spheroids are used as a control. Scale bar is indicated. (**D**) Representative Western blotting for eIF6 and AFP expression in spheroids. AFP protein levels are reduced in eIF6-silenced spheroids. (**E**) Spheroid area measurement expressed in mm^2^. Bold black lines indicate mean ± SD. Results of two-tailed *t* tests are shown. **** *p* value ≤ 0.0001. *n* = 7 spheroids/condition.

**Figure 5 ijms-23-07720-f005:**
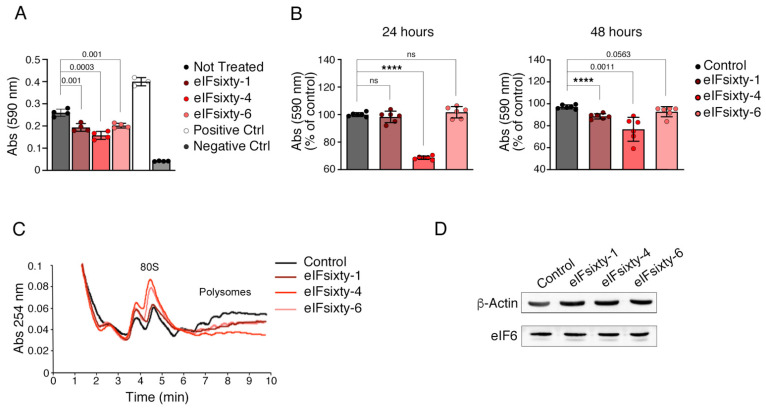
eIFsixty-i compounds inhibit eIF6-60S binding and impair the initiation of translation. (**A**) eIF6-60S binding is compromised in Huh7.5 cells upon eIFsixty-i treatment as shown by ELISA assay. The following concentrations were used: eIFsixty-1, 1.4 µM; eIFsixty-4, 5 μM; eIFsixty-6, 1 µM. Positive control: recombinant eIF6 protein plus pure 60S ribosomes. Negative control: pure 60S ribosomes only in the reaction. Data are representative of three independent experiments. Results of two-tailed *t* tests are shown. (**B**) Histograms show cell viability of Huh7.5 cells at 24 and 48 h after addition of each eIFsixty-i compound, as indicated in (**A**). Data are represented as percentage of control (untreated). Results of two-tailed *t* tests are shown. **** *p* value ≤ 0.0001; ns indicates not significant. (**C**) Representative polysome profile for Huh7.5 cells treated with each eIFsixty-i compound (eIFsixty-1, 1.4 µM; eIFsixty-4, 5 μM; eIFsixty-6, 1 µM). 80S and polysomes are indicated. Note that eIFsixty-i treatment impairs the initiation phase of translation, as shown by the increase in the 80S peak. (**D**) Western blotting: eIF6 protein levels are not reduced after 48 h of eIFsixty-i treatment.

**Figure 6 ijms-23-07720-f006:**
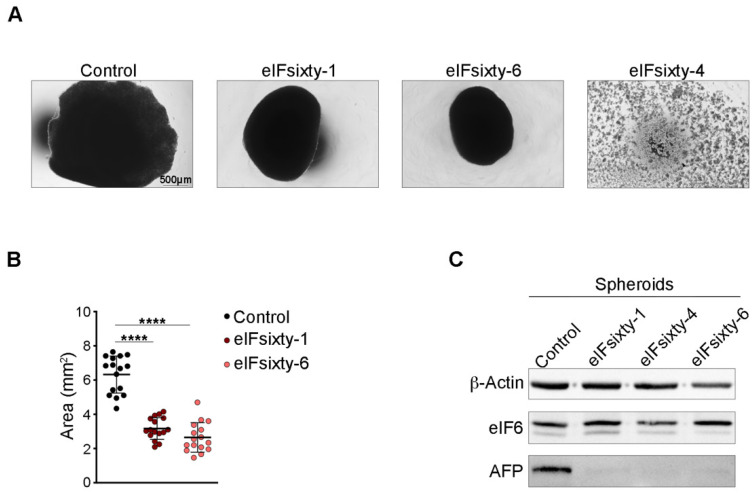
eIF6 pharmacological targeting impairs the growth of HCC spheroids. (**A**) Brightfield images of spheroids derived from Huh7.5 cells treated with eIFsixty-i compounds. The following concentration were used: eIFsixty-1, 1.4 µM; eIFsixty-4, 5 μM; eIFsixty-6, 1 µM. Scale bar is indicated. (**B**) Area of eIFsixty-1- and eIFsixty-6-treated spheroids is reduced compared to the untreated control. Area is expressed in mm^2^. Bold black line indicates mean ± SD. Each dot represents a single spheroid. Results of two-tailed *t* tests are shown. **** *p* value ≤ 0.0001. (**C**) Representative Western blotting of eIFsixty-i-treated spheroids for eIF6 and AFP. β-Actin was used as a loading control.

## Data Availability

Not applicable.

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
