# Peer review of "Inhibition of eIF6 Activity Reduces Hepatocellular Carcinoma Growth: An In Vivo and In Vitro Study"

_ijms, 2022, doi:10.3390/ijms23147720_

Round 1
Reviewer 1 Report
In the manuscript “Inhibition of eIF6 activity reduces hepatocellular carcinoma 2 growth, an in vivo and in vitro study 3”, Scagliola et al investigated the roles of eIF6 in HCC progression through in vivo and in vitro models. The authors demonstrated that eIF6 haploinsufficiency protects from HCC nodules development and liver injury in mice models. The eIF6 het mice livers show less fibrotic areas and proliferating cells compared to wild type. Knockdown of eIF6 significantly impairs HCC spheroids formation in Huh 7.5 cells. More interesting, the authors identified several eIF6 pharmacological inhibitors and found they can reduce the growth of HCC spheroids in vitro. The work is interesting, and the finding is appealing. However, the following revisions need to be considered before its consideration for publication.
The eIF6 pharmacological inhibitors used in this study are not available to the readers. The detailed information on the three inhibitors will help other research teams and researchers repeat the current experiment easily.
The effect of the eIF6 pharmacological inhibitors is only investigated in vitro, It would be more interesting if the in vivo experiments are also available. We cannot judge the potential research and clinical prospects due to the lack of basal information on those three inhibitors. Or at least a deep discussion of this aspect should be added based on the current research progresses and reports.
In the original-images file, the date listed in Figure 1C is not the real raw data. The authors should update it carefully.
Author Response
Thank you very much for your work. Please find enclosed our replies:
- The eIF6 pharmacological inhibitors used in this study are not available to the readers. The detailed information on the three inhibitors will help other research teams and researchers repeat the current experiment easily. Reply: The compounds are all available to the reader, since they have been isolated from publicly available chemical libraries. We have now specified in the Method section the supplier and the formula (paragraph 4.5), we apologize for the misunderstandings.
- The effect of the eIF6 pharmacological inhibitors is only investigated in vitro, It would be more interesting if the in vivo experiments are also available. We cannot judge the potential research and clinical prospects due to the lack of basal information on those three inhibitors. Or at least a deep discussion of this aspect should be added based on the current research progresses and reports. Reply: We start from the last paragraph. We have inserted a paragraph in the discussion on the present limitations of this study (line 247). The in vivo study will be certainly done for eIFsixty-6 that is not toxic in vitro, it has a very interesting biochemical activity, and has never been tested. However, we need to synthesize large amounts and therefore we are trying to collect enough funding for that. eIFsixty-4 has never been used in vivo, too. Our data suggest that it is a strong inhibitor of protein synthesis, and quite toxic, we doubt that it will be really without side effects. eIFsixty-1 (as we already said in the discussion for the sake of truth) is an antibiotic, known as clofazimine. It is (was) used for Mycobacteria, and poorly tolerated. The story of this drug traces to the golden era of antibiotics, it worked very well on Mycobacteria isolates, but it was very toxic and even immunosuppressive. Several mechanisms of actions have been proposed. This said, its in vivo toxicity has been established, and our data suggest that part of its effects can be associated to inhibition of eIF6 binding to 60S. To wrap it up, two of the three compounds give proof-of-concept that eIF6 activity can be inhibited, but are not likely to be able to be developed from the clinical perspective. One, eIFsixty-6 is peculiar and we want to test it as soon as possible.
- In the original-images file, the date listed in Figure 1C is not the real raw data. The authors should update it carefully. Reply: Done, we have added them in the Supplementary part and also better defined the experimental sample size.
Reviewer 2 Report
Introduction part (first paragraph) should be motivate; for example, why NASH-related HCC is unique?
Mice diet should be included clearly; what is the research diet here?
Please include in results and figures the concentration of the three compounds tested, the inhibition is dose dependent? The conclusion seems to show that idea
Please include some possible side effects of blocking completely the binding of eif6
Author Response
Thanks for the comments. Hereafter our reply.
- Introduction part (first paragraph) should be motivate; for example, why NASH-related HCC is unique? Reply: gene expression and epidemiological data suggest a difference in NASH-driven HCC, compared to, for instance viral-driven HCC [1]. We added info and revised the intro.
- Mice diet should be included clearly; what is the research diet here? Reply: the diet is now fully described at line 330. It is based on high fructose, 45% fat and hepatotoxic and mutagen insults (CCl4; DEN injection).
- Please include in results and figures the concentration of the three compounds tested, the inhibition is dose dependent? The conclusion seems to show that idea Reply: Concentrations added, example legend of Fig. 5 (please note, we have added the concentration only in the legend for simplicity). We used the dosage that is calculated on the Ki50 of the binding inhibition of eIF6 to 60S [2]. This assay is dose-dependent effect, and in the micromolar range [2]. Please note that the Ki is calculated versus the global 60S population, in vitro, but the compounds may target selective conformations of 60S-eIF6, in vivo. In this project, we have decided to work always at the Ki50 of eIF6-60S binding, in vitro. Unpublished data show that in viability assay increasing time/concentration of all compounds, as expected, is toxic.
- Please include some possible side effects of blocking completely the binding of eif6. Reply: we are currently not sure that eIFsixty-n can block all eIF6 binding to the 60S for the reasons described before, the possible existence of different 60S conformations. Binding of eIF6 to 60S requires a very large interface and the compounds are probably hitting specific conformations. This said, we have added a speculative paragraph in the discussion. The most obvious side effect may be, speculatively, the total block of 60S biogenesis. Off target effects are also possible, but they have not been discussed.
- Pinyol, R., S. Torrecilla, H. Wang, C. Montironi, M. Pique-Gili, M. Torres-Martin, L. Wei-Qiang, C. E. Willoughby, P. Ramadori, C. Andreu-Oller, P. Taik, Y. A. Lee, A. Moeini, J. Peix, S. Faure-Dupuy, T. Riedl, S. Schuehle, C. P. Oliveira, V. A. Alves, P. Boffetta, A. Lachenmayer, S. Roessler, B. Minguez, P. Schirmacher, J. F. Dufour, S. N. Thung, H. L. Reeves, F. J. Carrilho, C. Chang, A. V. Uzilov, M. Heikenwalder, A. Sanyal, S. L. Friedman, D. Sia, and J. M. Llovet. "Molecular Characterisation of Hepatocellular Carcinoma in Patients with Non-Alcoholic Steatohepatitis." J Hepatol 75, no. 4 (2021): 865-78.
- Pesce, E., A. Miluzio, L. Turcano, C. Minici, D. Cirino, P. Calamita, N. Manfrini, S. Oliveto, S. Ricciardi, R. Grifantini, M. Degano, A. Bresciani, and S. Biffo. "Discovery and Preliminary Characterization of Translational Modulators That Impair the Binding of Eif6 to 60s Ribosomal Subunits." Cells 9, no. 1 (2020).